



# SMEARcore – Modular data infrastructure for atmospheric measurement stations

Anton Rusanen[1], Kristo Hõrrak[2], Lauri R. Ahonen[1], Tuomo Nieminen[1,3], Pasi P. Aalto[1], Pasi Kolari[1], Markku Kulmala[1,4,5], Tuukka Petäjä[1,5] and Heikki Junninen[2].

[1] Institute for Atmospheric and Earth System Research (INAR) / Physics, Faculty of Science, University of Helsinki, Finland
[2] Institute of Physics, University of Tartu, Estonia
[3] Institute for Atmospheric and Earth System Research (INAR) / Forest Sciences, Faculty of Agriculture and Forestry, University of Helsinki, Finland
[4] Aerosol and Haze Laboratory, Beijing Advanced Innovation Center for Soft Matter Sciences and Engineering, Beijing University of Chemical Technology (BUCT), Beijing, China
[5] Joint International Research Laboratory of Atmospheric and Earth System Sciences, School of Atmospheric Sciences, Nanjing University, Nanjing, China

*Correspondence to*: Anton Rusanen (anton.rusanen@helsinki.fi)

**Abstract.** We present here the concept of the SMEARcore data infrastructure framework: a collection of modular programs and processing workflows intended for measurement stations and campaigns as a real-time data analysis and management platform. SMEARcore allows various new SMEAR stations (Station for Measuring Earth Surface – Atmosphere Relations) to be built consistently with existing ones and learn from pre-existing experience. The faster establishment of new measurements particularly from a data flow point of view allows those stations to directly benefit from our general experience and from further development of visualization and analysis codes. It also establishes robust data pipelines that allow easier diagnosis of problems. We show practical examples how SMEARcore is utilised at operational measurement stations. This work differs from earlier similar concepts, such as those utilized at stations within ACTRIS (Aerosols, Clouds, and Trace Gases Research Infrastructure) and ICOS (Integrated Carbon Observation System) networks, in two important aspects: by keeping all the processing under the control of the data owners and being extensible to new instruments.

## 1 Introduction

The volume of environmental data doubles faster than every two years (Guo, 2017). Atmospheric composition is continuously monitored with a combination of satellite remote sensing (e.g., Drusch et al. 2012; Beamish et al. (2020)) and comprehensive in-situ observations (e.g., Kulmala, 2018; Petäjä et al. 2020). The data is integrated and synthesized in a suite of Earth System models (e.g., Hurrell et al. (2013); Randall et al. (2019)). There are ambitions towards digital twin of the whole Earth System (Bauer et al. 2021), which would enable up-scaling, incorporation of human actions and taking advantage of advances in digital information technology to provide solutions towards sustainable future.



Managing the big data related to the atmosphere is a challenge. Here we place a focus on ground-based in-situ atmospheric observations. In this field, the recent technological advances and particularly a wide use of on-line atmospheric high resolution

mass spectrometry allow us to determine concentrations of trace gases and chemical composition of atmospheric aerosol particles in unprecedented accuracy (e.g., Junninen et al. 2010; Yao et al. 2018; Wang et al. 2020). At the same time, there is a constant need for observations at a higher spatial resolution and therefore more stations that provide targeted observations for the region e.g., to tackle specific issues related to air quality or climate change (Kulmala, 2018). With the number of stations and instruments, scales also the amount of available data. Modern atmospheric observations are not single observation points

but operated in a network providing harmonized and high-quality data (Laj et al. 2020). To take full advantage of these observations, it is imperative that these systems are well defined, documented and the measurements themselves need to be monitored for measurement instrumentation and hardware malfunctions and anomalies. Measurement systems must also be flexible and facilitate changes in the hardware, personnel, and software as they are inevitable in practice.

In the European scale, topic-specific research infrastructures have been set up to provide harmonized observations, such as Integrated Carbon Observation System (ICOS, Yver-Kwok et al. 2021) and Aerosols, Clouds, and Trace Gases Research Infrastructure (ACTRIS, e.g., Pandolfi et al. 2018). The global perspective is available through World Meteorological Organization's Global Atmospheric Watch (WMO-GAW, Laj et al. 2020). The comprehensive and co-located European infrastructures are endorsed by the SMEAR concept (Hari et al. 2016; Kulmala, 2018). Such stations can be tailored to tackle

different grand challenges, such as air quality (e.g., Liu et al. 2020) or climate change (Hari and Kulmala, 2005; Noe et al. 2015).

The various large-scale networks provide mutually different and network-specific standard operation procedures for the stations, ensuring harmonized end user data in their thematic context. However, these procedures are not necessarily well

connected with each other, and general overview is often missing. Furthermore, the journey from raw data to the data formats provided to end-users is often very labor intensive and done by different people for different instruments. Documenting the steps taken to process the raw data clearly and reproducibly is not simple. The traceability of data deteriorates further when it is used in scientific articles, where reproducibility is a known problem (Buck 2015). Simply put, there can be no reproducibility without proper documentation of what was done, and ad hoc data analysis often lacks such rigor.


In this work, we introduce SMEARcore, which aims to answer the problems of data management pertinent both for experimental campaigns and long-term measurement stations such as the SMEAR stations (Hari and Kulmala, 2005). It provides a set of modular tools for acquiring, transporting, indexing, monitoring, storing, and analyzing data. SMEARcore also provides a consistent interface to access the collected data and enables development of other programs on top of it or

embedding the results in, for example, webpages. We illustrate the key features of SMEARcore and show that a station running SMEARcore will provide a streamlined data pipeline from the instruments, measurement computers and databases all the way



to the end-user. SMEARcore features functionalities that provide the station managers with real-time updates on the data quality, data collection problems and status of the instruments, measurement computers and accessories. The system indexes this ancillary data, so that it can be accessed for further analysis. This indexing enables the implementation of routine

calculations, such as calibrations and visualizations, to be done automatically to aid operators to identify and solve problems with data collection. This standardization of operations and analysis allows us to do science faster, more reliably and there is a continuous process of supporting metadata and documentation generation for future reference.

SMEARcore has 4 main goals: Collect the raw data from disparate sources, monitor and display this process, provide access

to this raw data in a common format for further analysis, and do routine analysis to aid operators. We decided to make a modular architecture, which allows us to utilize already existing software solutions whenever possible. It also allows us to program on top of stable interfaces so that the modules are replaceable, and indeed our adjustments to different stations swap the implementation of modules. We hope that in the future this will also allow independent development of data infrastructures based on our interfaces.

## 2. Workflow

To effectively operate and expand a network of atmospheric stations, the observations need to be harmonized and supported by coherent data and document management. The purpose of these supporting actions is to keep the processes from observational raw data to data products as stream-lined and simple as possible. There seem to be no common tools for getting from measurements into well-structured data that would be widely used in the atmospheric sciences community. Thus, creating

a consistent set of processing tools for collecting and processing station data is necessary, and in some subsets of measurements this is already being done (e.g., Mammarella et al., 2016).

Data management is not only about checking that consistent calculations have been made. As with any system, errors can occur, for example: computers crash, power gets interrupted, networks are throttled, reagents run out, somebody forgets to run

a script or analyzer inlets foul. Some of these might cause problems for the measurements, some might just temporarily halt data transfers, but in any case, we need to know if something unexpected happened. For this to happen in a timely manner, parts of the analysis must be automated and monitored. If we need to wait weeks or months for a responsible person to analyze the data and notice a problem, we cannot intervene when it matters most, and useful data is lost. Same goes for transferring data out of the measurement computers, monitoring the state of those computers and backups. For these reasons, routine

operations should not be a manual process whenever it is possible to automate them.



What does one require from a station scale solution when systemically collecting observational data? The details vary slightly based on what one measures and what larger scale infrastructures the station belongs to, but in general the requirements can be summarized into six categories:

1. The ability to collect raw data from several measurement computers.
2. Performing routine analysis that combines several measurements, such as inversions.
3. Storing raw and derived data for an intermittent period for analysis and visualization
4. Displaying this data to the people conducting measurements for quality control
5. Transferring data to long term storage as backup and to vacate local space for new data.
6. Log what files are processed, how, when and present this metadata.

It should be noted that within SMEARcore we assume that the raw data for the analysis is provided by a combination of sensors and instruments, and the associated data acquisition software producing a raw data file. This means we leave making these raw data files up to the data acquisition software.

In practice, a SMEARcore installation is defined by a set of computations, defined as workflows, implementing these steps for a set of instruments and analyses and the backing computational infrastructure. We will now go through some of these workflows to explain what we mean by this concept. It should be noted that the individual workflows are simple since they are focused on a single purpose. Any complex analysis will utilize the results of previous workflows and the challenge is mostly in coordinating their execution. In practice one usually also includes checks to avoid duplicating work already done in previous workflows, but these are omitted in this paper for clarity. The underlying technical solutions will be described in later chapters.

## 2.1 Time series data

This is the simplest case of data processing in SMEARcore. The process is visualized in Fig. 1, which contains reading the raw data in and providing the data to visualization and long-term storage in different forms. The input data from different instruments differs mostly in how the instrument-specific raw data format should be interpreted and parsed for visualization. A practical example of such a data process is e.g., acquiring total sub-micron aerosol number concentration from a Condensation Particle Counter (e.g., Mordas et al., 2008). In a conceptual framework this dataset is a timeseries of a single parameter with native averaging from the instrument. The data streams include a timestamp, number concentration, information on the time resolution, and relevant metadata for the instrument and measurement location.

We often must create derived datasets from our raw data, for example to do calibrations or calculate new variables. This is accomplished in the workflow in Fig. 1 by having a node that operates on previously collected raw data. In practice plots and collecting several datasets for export in different formats also conform to this pattern, as they are just data products.





### 2.1.1 Examples of workflows


An example of producing derived datasets is the inversion of the raw datafiles produced by the Differential Mobility Particle Sizer (DMPS; for more details see e.g., Aalto et al., 2001 and Kulmala et al., 2012). The DMPS size selects aerosol particles based on their electrical mobility using differential mobility analyzer (DMA). The concentration of different sized particles is measured by condensation particle counter (CPC). Therefore, the primary data measured by DMPS is particle concentration

from CPC at different operating voltages of the DMA, and the voltages need to be converted into a size range. The auxiliary data (various flow rates, pressures and temperatures) which are needed for this inversion are usually stored in the same raw data file (for more details on the inversion process of DMPS raw data, see e.g., Kulmala et al., 2012). This means the required procedure is the workflow in Fig. 1, with the inversion function handling the processing node.

Another similar workflow is the processing of flux data from eddy-covariance (EC) measurements. The EC is a technique

which utilizes high-frequency measurements of wind and atmospheric variables (e.g., $CO_2$, $H_2O$ or particle concentrations) for calculating vertical turbulent fluxes between atmosphere and Earth surface. The collected datafiles are 10 Hz raw measurement data, and the EC flux is calculated from the covariance of the fluctuating components of vertical wind and the quantity of interest over some representative time window (typically 30 minutes). The EC data are further processed with the help of auxiliary meteorological data. The creation of derived data involves applying several data processing methods such as

detrending, despiking, coordinate rotation, dilution correction and covariance calculations (for detailed descriptions of these methods, see Mammarella et al., 2016). Most atmospheric data processing implemented within SMEARcore can be abstracted to such branching workflows

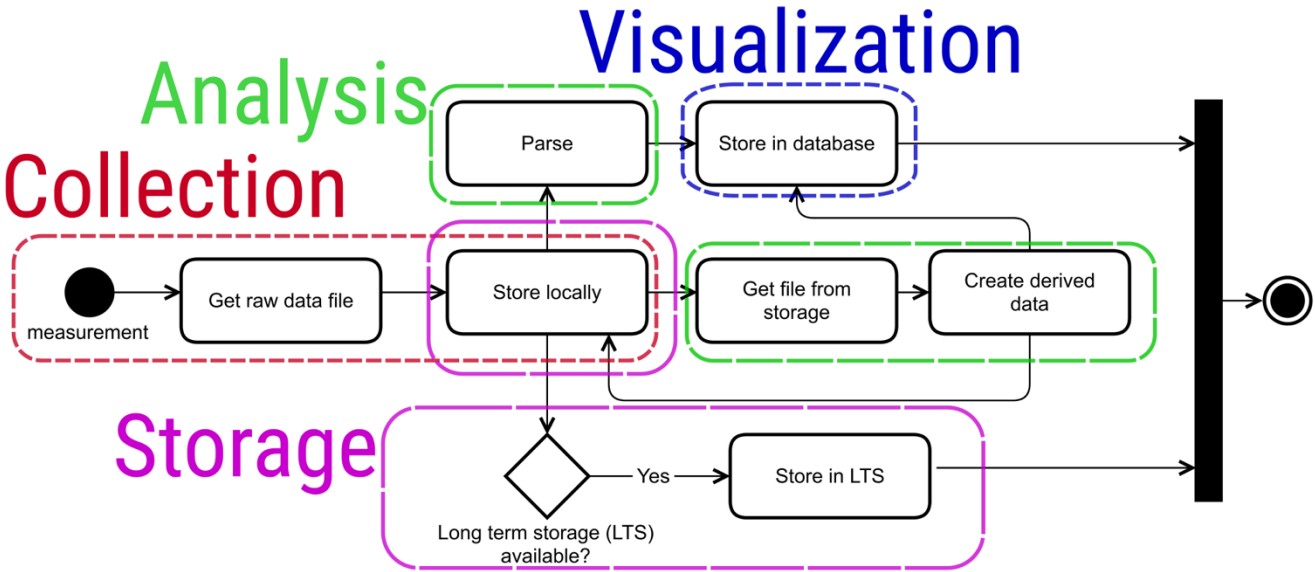



**Figure 1. Workflow of processing of a raw file containing time series data and creating derived dataproducts from it. The datafile is collected, parsed and stored, then various further processing can be done to it to create derived data. The different colored hashed boxes indicate what part of SMEARcore is involved in each processing step.**

### 2.1.2 Metadata and conventions

Any data files usually need metadata to be interpreted correctly. This is information such as measurement units, calibrations, column names in the data files etc. In our case we also produce metadata about the data processing itself: what files were processed when, how much data was there, what ancillary data was used in the processing and where the files can be located.

One file format used to solve this problem in infrastructures is NASA-AMES used by ACTRIS. There the file metadata is stored within the file itself as extensive header lines. In our case this would lead to extensive duplication of the data in many cases, and it is not appropriate for the metadata about the processing itself. Thus, we store metadata mostly as database tables and link to other files as necessary. Limited sets of metadata can be exported with the files by workflows to produce other formats.

### 3 Practical execution

In this section we describe the software & hardware used for SMEARcore. Any centralized solution needs at least three things: a server, storage space and a network connecting the computers. The server is responsible for the computing work involved in collecting, indexing, and serving the data to end users. It also provides the platform for any data analysis tasks defined by the workflows. An overview of the system is shown in Fig 2.

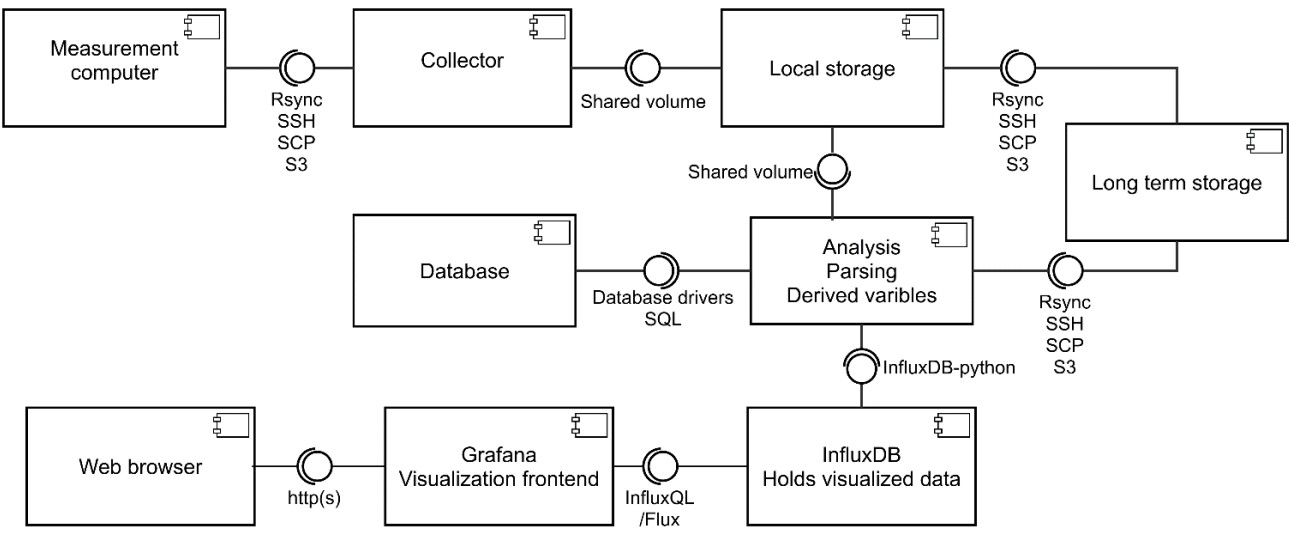



**Figure 2. A schematic picture of the different modules and interfaces between them in SMEARcore. The implementation within the box depends on the available hardware on the station, while the interface allows the implementation to change. For example, the analysis does not care where the data is stored if the interface allows retrieving it. The connections with the circle and cup represent the interfaces between the components. The labels in the interface refer to the technologies currently supported in at least one of our SMEARcore installations.**

## 3.1. Software

SMEARcore is built mostly with Python, which is a general-purpose programming language. The choice was based on the permissive license and availability of relevant libraries both for data handling as well as analysis. Python also allows one to call other programs via command line interfaces, which extends our available options by using analysis and processing codes written in other languages.

The entire system is packaged as docker containers, allowing easy installation to a single server via docker compose or on any Kubernetes enabled platform with a series of configuration files. Additional configuration is required to set secrets such as passwords and various routing options as well as selecting the different modular components of the system.

### 3.1.1 Storage

There are two kinds of storage we need. First is the local storage. This is used to store the files while they are being processed. The other is long-term storage which is used to save the files when they are not in active use. These may reside on the same device or service. Long-term storage may also be completely external such as another service or offline backups. These storages act as the repository of all the measurement data in the system.

We can use multiple backends from local disks to cloud based s3 storage. For local installations failure protection is a desirable feature, so setting up a raid and regular backup schedule is recommended.

### 3.1.2 Databases

The databases are used for monitoring the service itself, what files have been collected, what workflows have been run on which datafiles and what was produced. They also feed column form time series data to the plotting software. This means the databases store the status of the station.

We use three different databases: InfluxDB is a timeseries database that integrates with the visualization software and holds the time series data. PostgreSQL or MongoDB holds the information used to coordinate running the workflows. SQLite is used in some versions to hold information about which files have been processed, but this can also be done with the other two databases. The data itself is still stored in the original files and accessed on demand. This is due to a variety of formats, which are not suited for column storage, such as multidimensional fields.





### 3.1.3 Visualization

For online visualizations we are using Grafana (Grafana 2022). It provides a simple web interface with multiple views that the user can customize. This way it is possible to get both an overview of the health of the station as well as try to diagnose specific
problems by consulting the details from the interface. By default, views are configured to show the status of the measurement computers as measured by configurable monitoring scripts and the status of the data collection as indexed by the SMEARcore database. The interface also allows one to set alarm levels to get a quick notification of the station status.

### 3.1.4 Data collection

We have currently implemented four different data collection methods for different circumstances:

1. SSH access and rsync based transport. The primary option since it allows the server to control transport, but the measurement computer needs an SSH server and to allow incoming connections.
      2. SCP transport from the measurement computer with scripted WinSCP.
      3. Shared folders and scripted copying to them on the measurement computer.
      4. Storage into an s3 filesystem


Options 2, 3, 4 are similar since there the measurement computer is responsible for the transport and it is difficult to modify without physically going to the computer. They are sometimes necessary due to installation limitations. All forms of transport offer security via passwords or key-based authentication. They also allow us to restrict instruments so that they can access only their own data. This protects from unintentional data corruption and allows easy management in the case when instruments are
changed, or responsible people change.

### 3.1.5 Analysis

The analysis part of the software runs on top of Apache Airflow (Airflow 2022). This software allows us to define workflows such as calibrations and inversions based on multiple measurement files and run them on a schedule. The software also comes with visualizations of the states of the workflows and possible failures. The workflows themselves are composed of python
functions.

There is also an alternative implementation that is done by launching containers directly to do the analysis. In this case each workflow is self-contained.

### 3.2 Hardware

The server and storage can be co-located and thus far have been in our installations at Estonia (Noe et al. 2015), Beijing (Liu et al. 2020), and Arctic Ocean on board of RV Polarstern (AWI 2017) during the MOSAiC campaign (see e.g., Krumpen, et al., 2021). In these cases, both roles were fulfilled by a Network Attached Storage, NAS, system that supports container virtualization.





The hardware parts can also be distributed and run on any cloud provider or external server, allowing functionality without any extra hardware at the station. However, this solution requires a robust network connection from the station. This is the case with our SMEAR III (Järvi et al. 2009) installation, where we use a cloud platform provided by CSC (the Finnish IT Center for Science).

The choice of hardware boils down to how much processing needs to be done and how much data needs to be stored. It is also possible to separate the data collection and do postprocessing on any other platform as is traditionally done. There is, however, one part of the hardware that must be well planned, the local network infrastructure. As the network serves as the primary way of both transporting the data and representing the status, throttling or disruptions in the network result in degradation of the service. It is possible to run the system without access to the internet, but a local network is still necessary.

**4. Implementations**

**4.1 Case study: SMEAR Estonia**

The first installation was for the SMEAR station in Järvselja, Estonia (Noe et al. 2015). It uses a centralized server on location and rsync agents installed on the measurement computers so that data can be pulled by the server with data collection method 1 outlined in Section 3.1.4. SMEARcore software containers are run on this server on Docker and defined using Docker

Compose. The data collection and parsing workflows are organized as data acquisition units, DAQs, one pair for each monitored instrument type. The pairs are coordinated using RabbitMQ message queues and a MongoDB database for persistence. SMEARest offers visualization and metadata from the collection process through Grafana and direct access to the collected files using sftp and data transfer to off-site storage at University of Tartu high performance computing center.

At the SMEAR Estonia station there are currently 9 instruments which are monitored by SMEARcore at two locations, a main container, and a separate measurement cottage. The instruments measure aerosol properties, meteorological parameters, and background radiation. SMEARcore is integrated with mass spectrometer data analysis software tofTools (Junninen et al. 2010) and allows time-of-flight mass spectrometer data to be processed near-real time (once per hour). This makes it possible to present online processed data in the form of concentrations of chemical groups rather than intensities of single peaks.


The case study is from measurements in SMEAR Estonia station from period June 6[th] to June 7[th], 2021. The figures presented show screenshots from instrument specific real-time dashboards. For the case study, a warm period was chosen with daytime temperatures 24-26 °C and nighttime 10-13 °C. During this period, we see two daytime new particle formation (NPF) events and a night-time clustering event on 6.6.2021 8pm – 7.6.2021 2am. (Fig. 3 and Fig. 4). The NPF events are seen in the number-

size distributions measured by the Neutral cluster and Air Ion Spectrometer (NAIS) as formation of initially 2 nm air ions and



their subsequent growth to larger sizes during several hours (Fig. 3). The concentrations of sub-2 nm cluster ions measured by the Cluster Ion Counter (CIC) show diurnal variation, and the concentration of 3 nm ions shows a maximum during NPF events (Fig. 4). During these daytime new particle formation events the Chemical Ionization Atmospheric Pressure interface time-of-flight mass spectrometer (CI-APiTOF) did not observe any clear difference in the chemical composition of the sub-2 nm
clusters and aerosol particles from that typically observed on days without NPF occurrence (non-event days). However, during night-time clustering event CI-APiTOF observed a simultaneous increase both in highly oxidized organic molecules (HOM) dimer concentrations (Fig. 5) as well as in sulphuric acid concentration (Fig. 6). Similar night-time clustering events producing small (sub-10 nm) particles which do not grow into larger particles as in daytime NPF events have been observed also at SMEAR II station in Hyytiälä, Finland (Rose et al., 2018).




**Figure 3. On-line visualization of number size-distribution of positive (top) and negative (bottom) air ions measured with Neutral cluster and Air Ion Spectrometer (NAIS) at Järvselja SMEAR station on 6-7[th] of June 2021. Visualization is done with a custom plugin developed at the SMEAR station. The colormap and the number concentration scale can be changed by the user for their preferred viewing experience. The y-axis displays in logarithmic scale the ion mobility in units (sV/cm$^2$). Colorscale indicates the**
**number concentration (dN/dlog$_{10}$(dp), cm$^{-3}$).**

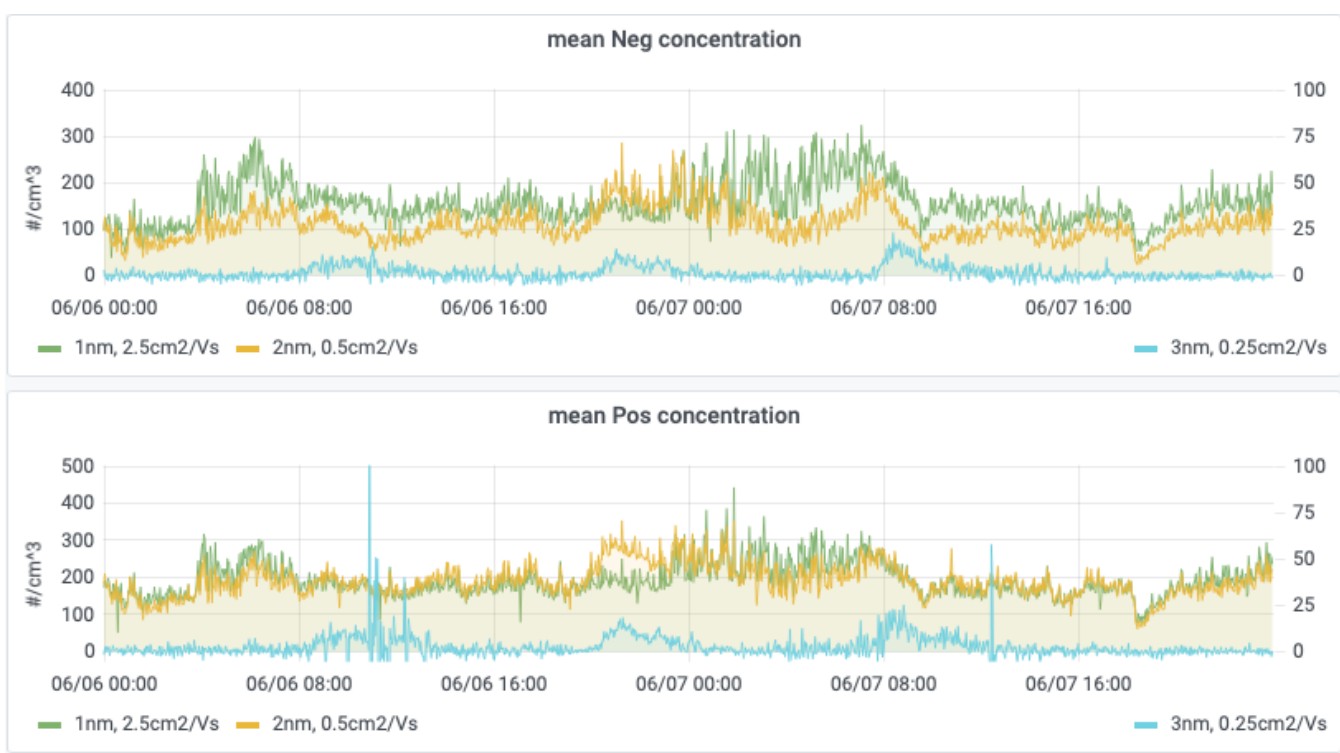

**Figure 4. Concentrations of atmospheric ions with geometric mean diameter of collection electrodes of 1 nm (green line) and 2 nm (yellow line) cluster ions, and 3 nm (cyan line) intermediate ions measured at the Järvselja SMEAR station during 6.6.2021 - 7.6.2021 measured by the Cluster Ion Counter (CIC). Due to the instrument design, the measured concentrations include ions from a wide mobility range. Negative ions are shown in the top panel and positive ions in the bottom panel.**


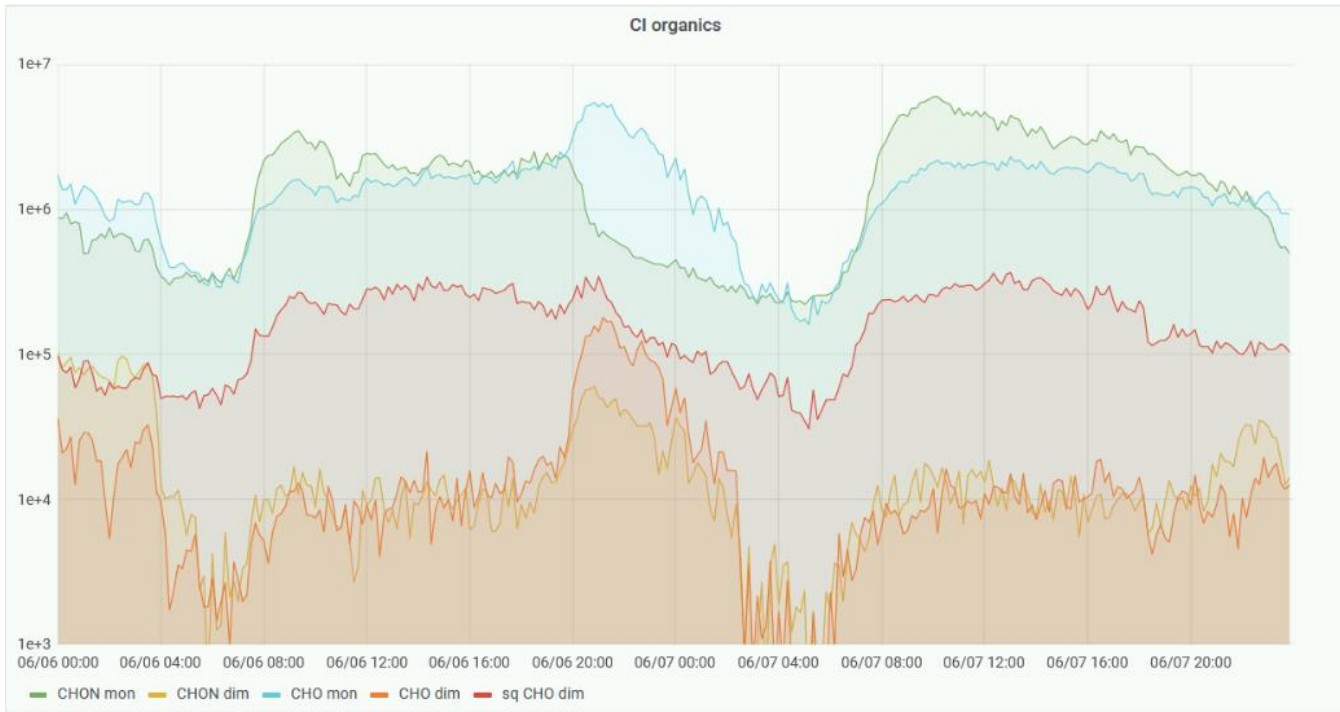

**Figure 5. Chemical Ionization Atmospheric Pressure interface Time-Of-Flight (CI-APiTOF) mass spectrometer on-line data of highly oxidized organic molecule (HOM) concentrations measured at the SMEAR Estonia station during 6.6.2021 - 7.6.2021. In the legend capital letters denote chemical elements present in the molecule, "mon" denotes monomers and "dim" dimers, prefix "sq" denotes that HOM are formed from sesquiterpenes, the others are formed from monoterpenes. Both sesquiterpenes and monoterpenes are volatile compounds emitted by vegetation.**





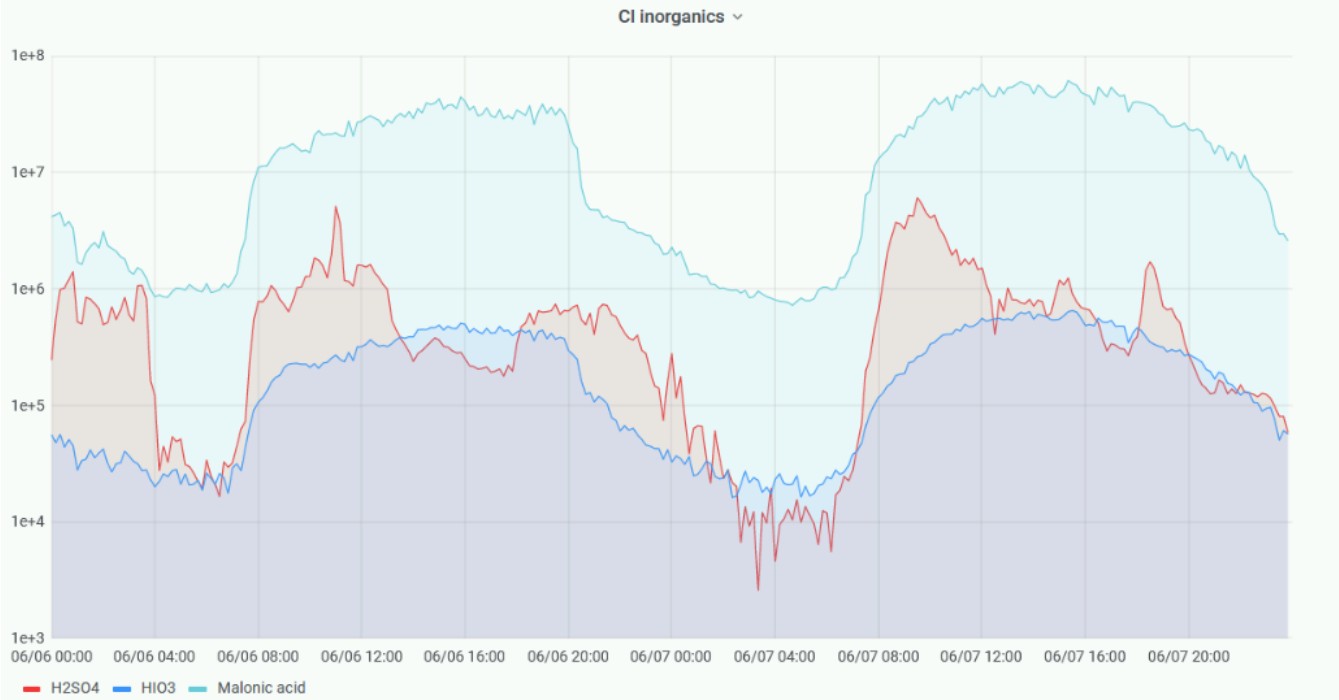

**Figure 6. CI-APi-TOF on-line inorganic acid concentrations (sulphuric acid shown in red, iodic acid in blue and malonic acid in cyan) measured at the SMEAR Estonia station during 6.6.2021- 7.6.2021. The formation of all three acids is initiated by solar radiation and formation of OH-radicals, however the precursor gases are different, $SO_2$, $I_2$, and a wide variety of volatile organic compounds, respectively.**

### 4.1.1 Utilizing Metadata

Figures 7 and 8 show dashboards on the instrument level and station level. As an example, an instrument dashboard from CI-APiTOF mass spectrometer is shown in Figure 7. For the instrument it is important that pressures in different chambers are in the correct range, too low pressure in the first chambers (SSQ and BSQ) indicate clogging of the orifice and mechanical cleaning is required, too high values again indicate malfunction of pumps and pump maintenances is required. In the dashboard monitoring values changes colors from magenta (too low) to green (correct) to red (too high). In addition to current parameter

readings also time series of pressure readings are displayed, this helps to identify the reasons for the problem and to see when exactly the problem surfaced. In the case of critical operational parameters an alert is given on the screen, but also an email is sent to the operator.

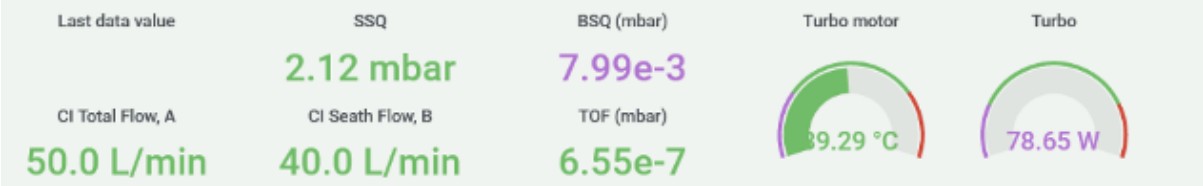





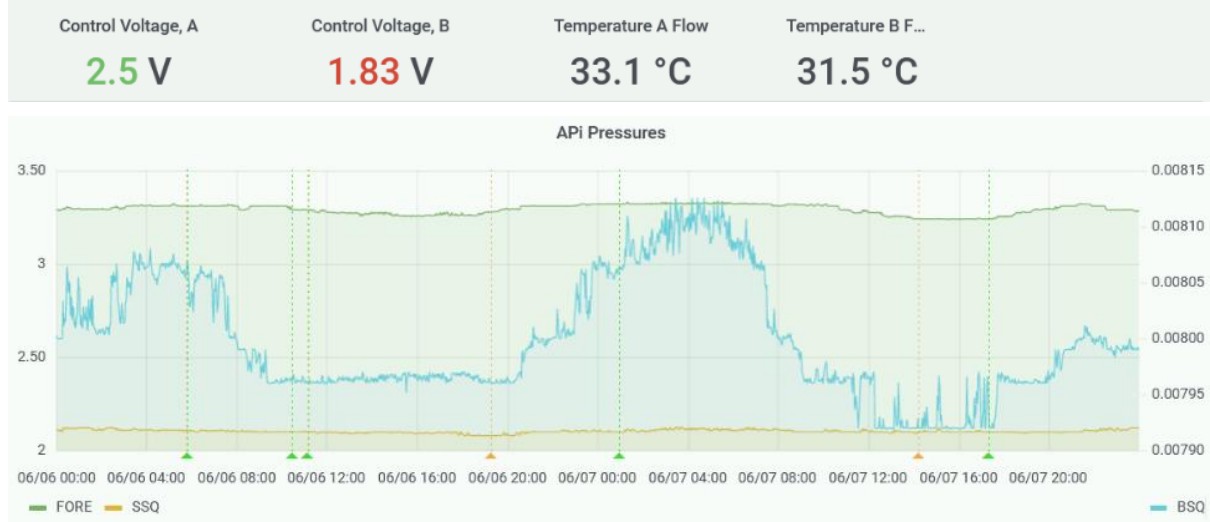


**Figure 7. Instrument monitoring parameters for the APi-TOF are plotted on-line and alerts are triggered if values are out of operation range. The orange and green arrows in the pressure graph indicate that an alert was present.**

Various auxiliary measurements can be constantly monitored to ensure the integrity of the measurement devices. Alerts call attention to abnormal readings and can be collected into figures such as Fig. 8. In this dashboard successful file readings are

marked with background color, but timing problems due to slow internet or intranet speeds are marked with green color, completely missing files are marked with red and the alerts are triggered. From Figure 8 is easily seen current problematic measurements, like GammaTracer and RADOS Cottage and also longer lasting problem like with TSI Flow Järvselja, this instrument is not at the station and thus the data is missing. When the instrument is fixed and brought back to the station the status will return to normal.






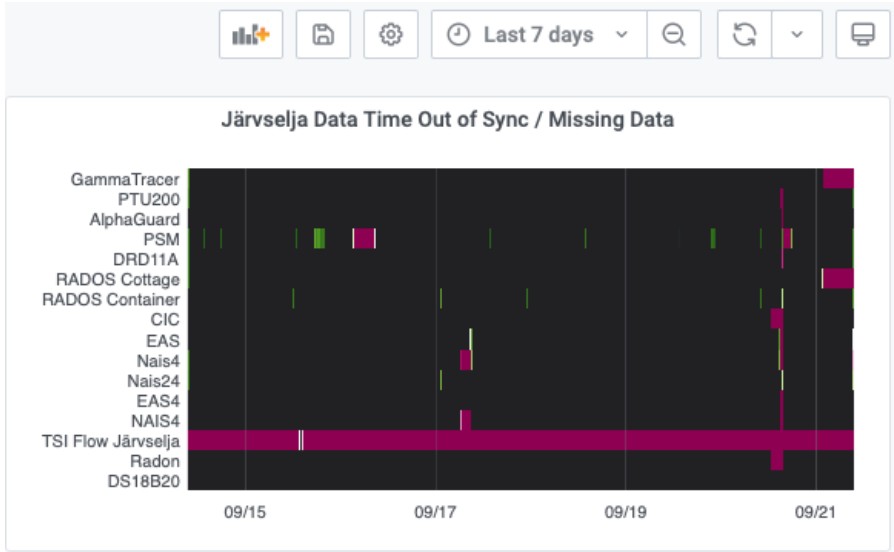

**Figure 8. Data syncing statistics from several instruments. Red and green indicate missing and mistimed data respectively. This allows detection of problems at the measurement and network.**


In addition to measurement metadata, we collect metadata from the data collection and parsing processes themselves. Last file access times, file sizes, count of parsed columns and how long the parsing took are all things that are monitored. Figure 9 shows an example of how this data can be used to determine the effect of a power cut on collected data. Input Voltage UPS indicates if UPS is being charged, if no input voltage, then system is operated from backup power. Gaps in graphs on 12th and

13th of January are result of such a long power cut, that communication with the station was also interrupted. File size is also smaller due to limited measurement time (Fig. 9).





**Figure 9. Effects of a long power cut on data collection at the SMEAR Estonia station on 10.-15.1.2021. UPS is fully discharged, and the station experienced network related issues due to no power. The observed data file size is considerably smaller due to missing data, and this is easily diagnosed from the collected metadata.**

### 4.2 Case Study: SMEAR III

Our newest installation is a parallel implementation of parts of the SMEAR III (Järvi et al. 2009) data analysis. The SMEAR III station is located at the University of Helsinki Kumpula campus in Helsinki, Finland. The instruments included are DMPS, basic meteorological instrumentation such as measurements of temperature, wind direction and wind speed, as well as trace gas measurement of ozone, carbon monoxide and nitrogen oxides. The storage backend is s3 in the CSC (IT Center for Science)





cloud service, with a computer simply pushing new files there as they are generated. The data is on a set retention policy, which means that old data is cleared automatically.

Visualization is done in Grafana, and the interface is facing the public internet, allowing the users access from predetermined remote locations. The analysis is done by running the workflows as graphs in Airflow. Figure 10 shows one such graph from the Airflow user interface. The analysis and visualization components are run in an OpenShift cloud service also at CSC. In this case the main design choice was enabling remote access to users, so the system could not be co-located with the measurements.

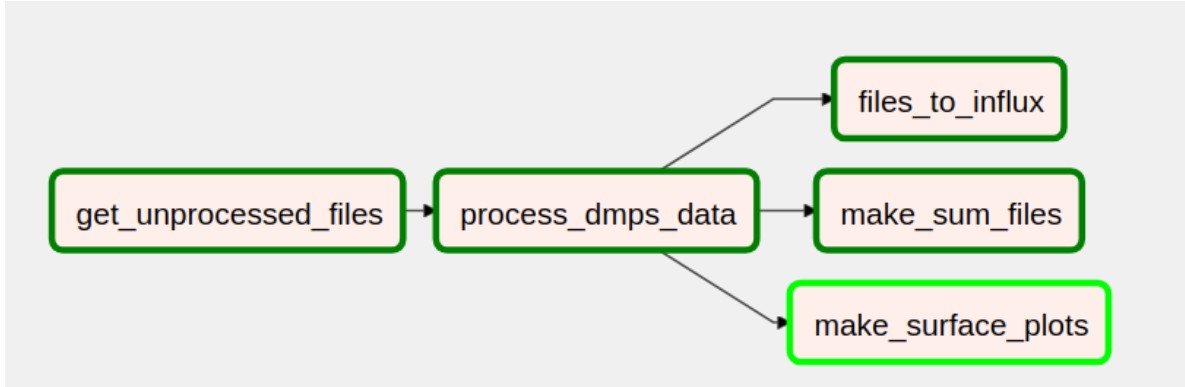


**Figure 10. The DMPS processing graph as seen from Airflow. It follows the same structure as the workflow in Figure 2. Influx refers to the database used for storing the processed data. Sum files are the processed file type and surface plots are used for visualization. The arrows represent dependencies, and the last three tasks can be done in parallel. All files are stored in a s3 instance. Colors represent the status of the task. In this case everything but make_surface_plots has completed successfully, while that task is still**
**running.**

Figure 11 shows an example visualization of the SMEAR III measurement data. On the morning of 18.2.2021 during 06:00-11:00 a clear surface temperature inversion is evident from the increase in temperature profile from 4 m to 32 m height above ground (Fig. 11a). At the same time, wind speed is also very low (below 1 m s$^{-1}$), and the wind is from north-to-northeast from the direction of nearby highway a few hundred meters from the measurement site (Fig. 11b, d). The temperature inversion and
the low wind speeds lead to inefficient mixing of the air close to ground and accumulation of pollutants emitted from vehicles in the morning traffic and other nearby anthropogenic sources. The particle concentration in the size range 3-820 nm and the concentration of nitrogen oxides (NOx, the sum of NO and NO$_2$) and carbon monoxide (CO) start increasing between 06:00-08:00, and at the same time the ozone concentration is depleted by more than a factor of 10 to below 1 ppb (Fig. 11c). Similar ground-level ozone depletion episodes have been observed at the Hyytiälä SMEAR II station mostly in autumn and winter
connected with low mixing layer, high relative humidity and low solar radiation intensity (Chen et al., 2018). The temperature inversion is strongest around 10:30, when the temperature measured at 32 m height (-10.9°C) is almost 5°C higher compared to temperature at 4 meters (-15.8°C). At this time are observed the highest particle concentrations (above 5•10$^4$ cm$^{-3}$) and NOx concentrations (above 200 ppb). The ozone concentration has already started to increase. By 11:30 the temperature inversion



has disappeared, the wind speed has increased, and as the result of more efficient vertical and horizontal mixing of the air the

measured pollutant concentrations are close to their typical background levels for the Helsinki area. This case study demonstrates how easy-to-use data visualization tools, which allow efficient comparisons between datasets from multiple instruments, can help in identification of interesting phenomena in the measurements.





**Figure 11.** An example of measurements of temperature at heights of 4 m and 32 m (a), wind direction (b), trace gas (ozone, nitrogen oxides and carbon monoxide) and particle concentrations (total concentration in size range 3-820 nm) (c), and wind speed (d) at SMEAR III station in Helsinki, Finland, on 18.2.2021.

## 5. Conclusions & outlook

We present a concept for station data management and acquisition using interchangeable components. The concept is in operational use at SMEAR Estonia and has been tested at several campaigns. The components we use are built on popular, well-known open-source projects. This framework is suggested for use at new SMEAR stations and could be useful for larger campaigns as well. Since our system is completely modular, different configurations allow it to be adapted for most common use cases. The system can also easily be extended for more instruments and in the future new technological solutions, as necessary. Compared to centralized solutions such as ICOS or ACTRIS stations, this allows the users to fully control how their data is processed, monitor it in real-time and control how it is transferred outside the station. While using our framework does require some technical planning to ensure sufficient hardware resources, we believe the benefits and possibilities of automated data analysis outweigh the costs. We show in two case studies how continuous visualizations of the data and metadata, such as instrument diagnostics and datafile availability, can help quickly spot interesting phenomena and abnormal situations in the measurements.

Since the SMEARcore software allows one to combine multiple data sources, it also provides new opportunities for networking measurement stations together and automatically cross-referencing diverse sources of data in routine operation of the station. An improvement for the management of measurements would be shared storage between stations, where one could check instrument settings or normal operating values at different stations. Another possibility for improving the data usage would be automatically integrating model or satellite data into the analysis or automatically producing the input files for such models, since they can be considered just data products in the SMEARcore framework. In short, automating data processing in the way SMEARcore does also provides opportunities to automate further steps of the scientific process.

## 6. Author contributions

AR, MK, TP, HJ, PA and PK participated in the initial design of the SMEARcore concept.

MK, TP and HJ participated in funding acquisition, resource acquisition and supervision of the project.

AR, KH, HJ, PA and LA participated in software development and data curation.

AR, KH and HJ investigated the concept by setting up and operating installations.

AR, KH, HJ and TN made the analysis presented in the examples and provided visualizations.

All co-authors participated in the writing and commenting of the manuscript.



## 7. Competing interests

The authors declare that they have no conflict of interest.

## 8. Acknowledgements


We acknowledge the following projects: ACCC Flagship funded by the Academy of Finland grant number 337549; Academy professorship funded by the Academy of Finland (grant no. 302958); Academy of Finland projects no. 1325656, 316114 and 325647; "Quantifying carbon sink, CarbonSink+ and their interaction with air quality" INAR project funded by Jane and Aatos Erkko Foundation; European Research Council (ERC) project ATM-GTP Contract No. 742206; and the Arena for the gap

analysis of the existing Arctic Science Co-Operations (AASCO) funded by Prince Albert Foundation Contract No 2859.

Technical and scientific staff in Järvselja, Beijing and Hyytiälä stations are acknowledged.

We thank Marjut Kaukolehto for discussions during the planning of SMEARcore.

This work was supported by European Regional Development Fund (MOBTT42), Estonian Research Council (project PRG714) Estonian Environmental Observatory (KKOBS, project 2014-2020.4.01.20-0281), Academy of Finland (grant no.

415  311932)

The authors wish to acknowledge CSC – IT Center for Science, Finland, for computational resources.

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
