# Peer review of "SMEARcore – Modular data infrastructure for atmospheric measurement stations"

_Atmospheric Measurement Techniques, 2022_

## Referee Comment (RC2)

**Journal: Atmospheric Measurement Techniques**

**Title**: SMEARcore – Modular data infrastructure for atmospheric measurement stations

**Abstract**: It is adequate; nevertheless, the authors should highlight why the differences with other infrastructures are important within the conceptual frame and not just mention them.

**1. Introduction**: The authors follow a common thread that is easy to follow. They correctly pose the conceptual problem within data processing in air quality networks and provide an overview to potential readers on reasonably similar current structures.

**Comment on the phrase:** 'To effectively operate and expand a network of atmospheric stations, the observations need to be harmonized and supported by coherent data and document management'. The observations or measurements recorded by air quality networks should be tested or validated but not harmonized. The measurement techniques used for monitoring air quality status should be harmonized to offer traceability of monitored air pollutants data.

**Figure 1** is explained in section 3, making it challenging to understand at first glance. The authors should provide information concerning Figure 1 in section 2.1.1.

**Figure 3**. The legend should be finished.

It would be of interest whether the authors could establish a cost-benefit relation between the centralized solutions such as ICOS or ACTRIS stations and the new conceptual framework.

The English editing should be notably improved.

---

## Author Comment (AC1)

**Response to referee 1 ( https://doi.org/10.5194/amt-2022-67-RC1 )**

We thank the reviewer for the comprehensive comments on our manuscript. Below we give our detailed response to each of the comments (shown in italics) and the changes or additions made to the manuscript based on them.

*However the manuscript needs some substantial rewriting. The general plan is not presented clearly enough.*
After the changes detailed in the responses below the general plan should be clearer.

*The abstract states that the processing is under the control of the data owners with a focus on the station level but later it is mentioned that data postprocessing can be done outside the station, and in the conclusion the authors mention multiple stations supervised by the same people. Data harmonization is emphasized which is a very good idea, but is that case the data owner can not really do whatever she/he wants anymore. Data harmonization is a key concept in networks like ICOS where the data processing is centralized leading to the harmonization. The SMEAR concept endorses European infrastructures like ICOS but the authors claim their concept to be different ; however in the conclusion they suggest networking stations together, cross-referencing their data and sharing storage between the stations.*
*While it is true that higher level networks require you to follow their protocols, many stations/campaigns have measurements that fall outside their scope. In these cases, sharing resources/protocols between such non-infrastructure stations is possible, which is why it was mentioned. SMEARcore is not a replacement for ICOS/ACTRIS protocols and we don't mean to imply it is.*

We added clarifying sentences to abstract:

"Secondly, by providing tools for making data interoperable in general instead of harmonizing a particular set of instruments"

"As such it is not meant as a replacement for these infrastructures, but to bring structured data curation to more measurements not covered by them."

We added to Introduction:

", in this paper meaning any process from raw data to products such as end-user data or diagnostics," *to clarify what we mean by analysis. Removed the words "ad hoc" to remove possible confusion.*

"Developing documented workflows for situations not covered by large-scale network protocols is a problem many stations need to solve."

"The interfaces also enable building small networks on top of SMEARcore directly."

"SMEARcore provides flexible and scalable framework that can be applied at instrument, station or multiple station level."

*Deleted the following sentence fragment from Workflow:*

"and what larger scale infrastructures the station belongs to"

*We also added clarifying sentences to conclusions:*

"This makes it useful for measurements not controlled by the centralized solutions."

"This means it is possible to establish smaller networks more easily with the software."

*For example, many measurement campaigns and SMEAR stations have lots of similar measurements which could easily be processed by same codes, cross-referenced if there are problems and stored in the same location, even if they are not part of any large-scale network. Since the SMEARcore could be ran in an internet accessible location, it would even be possible to define a "virtual station" that consists of instruments that are nowhere near each other.*

Another point which needs clarification is the use of SMEARCore in the frame of campaigns as it needs some hardware resources.

*There is chapter 3.2 about hardware. We added sentences to clarify the sensible minimum requirements (a reasonably modern computer, enough disk space and wired connections between it and measurement computers.):*

"In practice this usually means setting up routers and wired ethernet connection between the computers."

"In general any computer that supports container virtualization and has enough storage can work as the server"

Specific comments:

L18:why a faster installation of new measurement will allow a station to benefit from the experience of SMEARCore ?

*We changed faster to structured. Speed is an effect not a cause.*

L 33-39: the paragraph focuses on important amount of data and big data but this is not related to the accuracy of mass spectrometry. The raise of number of stations implies indeed more data but not at a single station and SMEARCore focuses on the station level.

*We added the sentence:* "Doing as much processing as possible at the station can aid in the management of the volume of data."

L 55: big infrastructures are by default not thought to be interconnected with each other, it is a plus when they do. Presenting the lack of coordination between them as a default is not correct. It is true that processing data can be labor intensive and lack documentation, mentioning this to highlight SMEARCore features is fair but saying that the large infrastructures do not automatize, trace and document their process is incorrect.

*We removed the sentence saying that they are not connected. Added sentence:* "Developing documented workflows for situations not covered by large-scale network protocols is a problem many stations need to solve." *The aim, as said, is not to replace any big infrastructures, but to aid in general measurements not covered by them.*

L 101: point two is more a plus than a default requirement.
*That ability is pretty much central for much of the existence of SMEARcore. Without analysis capabilities the station cannot effectively monitor itself or the instruments. Now, it is debatable as what counts as "analysis", here we have gone with "can do things with the files it collects".*

L 121: indeed but there is no conclusion related to SMEARCore features.
*We clarified the implication by adding to end of sentence* ", meaning that most instruments can be handled by SMEARcore almost identically."

L 123: « in a conceptual framework » would not be better said as « conceptually »?
Yes, we changed this.

L 138: the required procedure corresponds to the workflow.
We corrected this.

L 139: one workflow or branching workflows?
*EC would be a similar parallel, independent workflow. There is no branching here.* We added the word "independent" to clarify.

L 150: figure 1 legend, are each box a workflow ? If yes it would be to mention it.
*No, they are not workflows in themselves. Each one is a processing step in the workflow. The coloured division boxes are architectural units. We added the sentence to the caption:* "The black bordered boxes are steps in the workflow."

L 194: column form time series data is not clear.
*Any data of the format (timestamp, datapoint) is columnar form when multiple datapoints are present. We removed the confusing term column form.*

L 200: maybe « raw data » will be more clear than « data itself »
We changed this.

L 204: « This way it is ... » is not clear, something like « Multiple views allow to get » may be easier to understand.
We changed this.

L 219: « restrict instruments  so they can access only their own data.» is not clear.
*We extended and reformulated the sentence to:* "They also allow us to isolate instruments so that each has their own folders, and they cannot even accidentally overwrite other data."

*So, instrument A can write their files even if instrument B has the same name of raw files.*

L 222:Apache Airflow is only use for the analysis workflows, what about the others?
*Analysis is the only part with complex workflows. The data collection operates on a simple loop. Nothing in principle stops one from using Airflow for other parts, which would be a possible future development.*

L 252-253: « SMEARest » SMEAR Estonia? Grafana processes the metadata? The whole sentence is not really clear.

*We split the sentence into two parts. Grafana offers visualizations (as normal), as well as some additional metadata about the collection process itself. It does no processing by default (it's possible to define functions, but that is beside the point).*

L 262-274: too long for the purpose of the manuscript.

*We removed unnecessary details. The removal of figure 6 and supporting sentences also shortens this section. The purpose is to show how having access to the data of multiple instruments helps in interpreting the situation.*

Figure 3 legend: the information of the custom plugin is interesting and it would be more appropriate to move it to the core of the manuscript.

*We moved it to the chapter discussing the plot.*

Figure 5 and 6 are redundant, one would be enough with the legend specifying that inorganic data can be presented the same way.

*We removed previous Figure 6 (adjusted other figure numbers and fixed figure labels in text) and added a note about inorganic data in the caption.*

L 306: remove « but », an email is also sent ot the operator.

We replaced ",but also" with "and".

L 318: remark on SMEARCore, it should trace the removal of the instruments from the station in order not to display false status.

*It's impossible to resolve a missing instrument without operator given metadata. In this case one can also just remove the instruments from the workflows/views. So, this is not really something SMEARcore can trace automatically.*

Figure 8: may be it can indicate how far in the pas the data are available.

*That is a possible improvement that was not implemented during this study.*

L 338: « parallel implementation » is not very clear.

*The station operated normally at the same time; we did not replace any functions. Instead, we had our own analysis pipeline. We removed the word parallel, since this is not a necessary detail for the manuscript.*

L 346: « running the workflows as graphs in Airflow » sentence is a bit to technical. Maybe something like « the workflows in Airflow are defined by graphs ».

We changed it to: "The analysis workflows in Airflow are defined by graphs"

---

## Author Comment (AC2)

**Response to referee 2  https://doi.org/10.5194/amt-2022-67-RC2**

We thank the reviewer for the comprehensive comments on our manuscript. Below we give our detailed response to each of the comments (shown in italics) and the changes or additions made to the manuscript based on them.

Broadly, the subject reported in the manuscript is interesting and may have practical applications within the environmental field, particularly in air quality management. The description of the suggested concept could develop better to offer an easier comprehension to potential non-expert readers.

*We clarified the concept at several points. We have reiterated that this concept is for a set of common tools to be used by stations for currently not harmonized workflows.*

*We added sentence to introduction:* "Developing documented workflows for situations not covered by large-scale network protocols is a problem many stations need to solve."

*We removed the phrase:* "and what larger scale infrastructures the station belongs to," from *chapter two.*

*We added a sentence:* "In general, any computer that supports container virtualization and has enough storage can work as the server."

*This is to highlight that we are talking about station scale solutions here.*

*We added a sentence to conclusions:* "This makes it useful for measurements not controlled by the centralized solutions."

*In addition to change in abstract below.*

**Several Figures should be improved to facilitate understanding.**

*We removed Figure 6 as unnecessary. We revised Figure 1, 3 and 4 captions for clarity.*

**The English editing should be notably enhanced./ The English editing should be notably improved.**

*Some excessively long sentences were split into several sentences. Many word choices were revised to simpler alternatives. Unnecessary words were removed from several places.*

**Comments from supplement:**

Abstract: It is adequate; nevertheless, the authors should highlight why the differences with other infrastructures are important within the conceptual frame and not just mention them.

*We added sentences to abstract:*

"Secondly, by providing tools for making data interoperable in general instead of harmonizing a particular set of instruments"

"As such it is not meant as a replacement for these infrastructures, but to bring structured data analysis to more measurements not covered by them."

1. Introduction: The authors follow a common thread that is easy to follow. They correctly pose the conceptual problem within data processing in air quality networks and provide an overview to potential readers on reasonably similar current structures.
*The comment requires no action.*

Comment on the phrase: 'To effectively operate and expand a network of atmospheric stations, the observations need to be harmonized and supported by coherent data and document management'. The observations or measurements recorded by air quality networks should be tested or validated but not harmonized. The measurement techniques used for monitoring air quality status should be harmonized to offer traceability of monitored air pollutants data.
*We removed the loaded word harmonized. Supporting measurements with data workflows was the point of the sentence.*

Figure 1 is explained in section 3, making it challenging to understand at first glance. The authors should provide information concerning Figure 1 in section 2.1.1.
*We reference Figure 1 and explain the workflow already in section 2.1. In section 2.1.1 we refer to parts of the workflow in an example manner. We explain the colored categorizations are explained in section 3. The figure serves several purposes, only one of which is relevant at a time in each chapter.*

*We added text:* "The different colored hashed boxes indicate which implementation part of SMEARcore is involved in each processing step. The implementation parts are explained in Section 3." *to the caption.*

*We added text:* "Section 3 explains how the various parts are implemented in SMEARcore." *into section 2.1.1.*

*We added a sentence:* "The black bordered boxes are steps in the workflow." *to the caption.*

Figure 3. The legend should be finished.
*The "Par..." in the legend is simply the interface cutting off the phrase "Particle number concentration", since that column has been resized smaller than the text. It's not ideal, but this is what it looks like in normal use which is what is being demonstrated by the figure. The meaning of the colorscale (which we assume is what is meant by legend here) and the y-axis is explained in the figure caption.*

It would be of interest whether the authors could establish a cost-benefit relation between the centralized solutions such as ICOS or ACTRIS stations and the new conceptual framework.
*This would require considerable extra research and as such is not feasible in the scope of this article.*

---

## Editor Decision (ED1)

Dear authors,

Thank you very much for the revision of the manuscript. The content of this paper is very valuable for the community, which was pointed out by the two referees too. For this reason, we should try to bring the publication up to the necessary standard.

General:

The current version still requires further improvements.

A further revision of the English language is crucial.

 (see e.g. suggestion: line 17, Conclusion, line 385)

All abbreviations that you need must be written out and/or explained.

It is also necessary to check all punctuations, various commas are missing.

The paper should better highlight the functionalities and advantages of SMEARcore and explain them using the examples. Often we only find a description of the example, needs to be improved.

(A good examples are the descriptions of Figure 6 and 7; the description of Figures 3, 4 and 5 needs to be improved to describe the advantage of SMEAR's functionality).

You will find additional comments below. From my point of view, it is also crucial that the SMEARcore can also bring additional benefits in combination with a central network. This should be stated somewhere.

Abstract:

Revision of the Abstract (English and content) is required, examples see below.

Content: e.g. SMAER can also be an asset in parallel to a central solution, see my comment in the Conclusions).

English: the following is no understandable for experts not experienced in this filed.
line 17:
SMEARcore allows new SMEAR stations (Station for Measuring Earth Surface – Atmosphere Relations) to be built consistently with existing ones and to utilize pre-existing experience in data curation.

Suggestion: SMEARcore enables new SMEAR (Station for Measuring Earth Surface - Atmosphere Relations) stations to be integrated in a way that is consistent with existing stations and transfers existing data curation experience to the new station.

Introduction:
Line 74-75: English revision
Line 94-101: English revision

.

Line 160, capture figure 1

Include explanation of the black dot with circle, on the right of the figure in figure capture.

Line 163-171: English revision

Line 180: Figure description has to include all abbreviations used in the figure

Conclusions, line 382:

This makes it useful for measurements that are not controlled by centralized solutions

Add: or it can be used as a backup of the data owner in parallel with data transfer to centralized networks.

Conclusions, line 385.

Your current text:

We show in two case studies how continuous visualizations of the data and metadata, such as instrument diagnostics and datafile availability, can help quickly spot interesting phenomena and abnormal situations in the measurements.

Suggestion:

We demonstrated with two case studies how continuous visualizations of data and metadata, such as device diagnostics and data file availability, can help to quickly identify interesting phenomena and abnormal situations in measurements.

Thanks in advance for taking this in account, kind regards Brigitte

---

## Author Response (AR2)

**Response to editor comments**

Thank you for the comments. Below, we go through them and the corresponding changes in the manuscript. Line numbers in brackets "[]" indicate lines in the markup version of the document.

**The current version still requires further improvements.**

**A further revision of the English language is crucial.**

We revised several sentences. Various overly elaborate verbs such as "utilize" were replaced with simpler alternatives. [17-21, 23, 24-26, 46-47, 70, 75-79, 85, 96-107, 109-110, 116, 119, 122, 124-127, 130-131, 133, 135, 140, 153, 156, 163-165, 170-186, 188-189, 203, 209, 212, 220, 223, 226, 231-232, 238-242, 244-245, 272, 354-355, 365, 378, 381, 391-392, 395, 409-413]

**All abbreviations that you need must be written out and/or explained.**

We spelled out all remaining abbreviations [184,197-200,259-260,283,354,433]. We also added explanations of various technical terms. [197-200, 224-225, 240, 283, 354]

**It is also necessary to check all punctuations, various commas are missing.**

We added commas to [26, 46, 119, 347]. Simplifications were made to several sentences, reducing the need for commas [43-44, 67, 109-110, 116, 122, 126, 130]. We checked the punctuation of the text with the help of language tools in Word.

**The paper should better highlight the functionalities and advantages of SMEARcore and explain them using the examples. Often we only find a description of the example, needs to be improved. (A good examples are the descriptions of Figure 6 and 7; the description of Figures 3, 4 and 5 needs to be improved to describe the advantage of SMEAR's functionality).**

Added points about how the interactive and linked plots are an advantage. Also, the plots depend on data processed by SMEARcore. [307-308, 314-315, 321-323] Note that plots 3, 4, 5, do represent typical plots in this field, so the main difference any normal plotting is that they are automatically generated for the time periods in question. They are needed to show that we can detect the event based on a dashboard that has all three, but the plots themselves are not exceptional.

**From my point of view, it is also crucial that the SMEARcore can also bring additional benefits in combination with a central network. This should be stated somewhere.**

This is now mentioned in the abstract and the conclusions based on the other comments. [27, 409-410]

**Revision of the Abstract (English and content) is required, examples see below.**

The abstract was simplified, and the content additions below were added.

**Content: e.g. SMAER can also be an asset in parallel to a central solution, see my comment in the Conclusions).**

We added "be used in addition to them and to" to the last sentence of the abstract.

English: the following is no understandable for experts not experienced in this filed.

line 17: SMEARcore allows new SMEAR stations (Station for Measuring Earth Surface – Atmosphere Relations) to be built consistently with existing ones and to utilize pre-existing experience in data curation.

Suggestion: SMEARcore enables new SMEAR (Station for Measuring Earth Surface - Atmosphere Relations) stations to be integrated in a way that is consistent with existing stations and transfers existing data curation experience to the new station.

We changed the sentence as suggested.

**Line 74-75: English revision**

We changed the sentences from:

"The system indexes this ancillary data, so that it can be accessed for further analysis. This indexing enables the implementation of routine calculations, such as calibrations and visualizations, to be done automatically to aid operators to identify and solve problems with data collection."

To

"We store this metadata for further analysis. It allows us to automatically make calibrations and visualizations, which aid in identifying and solving problems with data collection."

**Line 94-101: English revision**

We simplified the paragraph from:

"Data management is not only about checking that consistent calculations have been made. As with any system, errors can occur, for example: computers crash, power gets interrupted, networks are throttled, reagents run out, somebody forgets to run a script or analyzer inlets foul. Some of these might cause problems for the measurements, some might just temporarily halt data transfers, but in any case, we need to know something unexpected happened. For this to happen in a timely manner, parts of the analysis must be automated and monitored. If we need to wait weeks or months for a responsible person to analyze the data and notice a problem, we cannot intervene when it matters most, and useful data is lost. Same goes for transferring data out of the measurement computers, monitoring the state of those computers and backups. For these reasons, routine operations should not be a manual process whenever it is possible to automate them."

To

"Data management is more than making consistent calculations. Any system can experience errors and a typical measurement station has many things that can go wrong. If errors are not detected, it is not possible to intervene in time and data is lost. It is critical to detect errors promptly and that requires automation of the detection. This monitoring should watch the data transfers, computing hardware, and calculations in addition to the measurements."

**Line 160, capture figure 1**

**Include explanation of the black dot with circle, on the right of the figure in figure capture.**

We explained the end point of the process in the figure caption.

**Line 163-171: English revision**

We changed the section from:

"Any data files usually need metadata to be interpreted correctly. This is information such as measurement units, calibrations, column names in the data files etc. In our case we also produce metadata about the data processing itself: what files were processed when, how much data was there, what ancillary data was used in the processing and where the files can be located.

One file format used to solve this problem in infrastructures is NASA-AMES used by ACTRIS. There the file metadata is stored within the file itself as extensive header lines. In our case this would lead to extensive duplication of the data in many cases, and it is not appropriate for the metadata about the processing itself. Thus, we store metadata mostly as database tables and link to other files as necessary. Limited sets of metadata can be exported with the files by workflows to produce other formats."

To

"Metadata is used to interpret data. Metadata consists of information such as measurement units, calibrations, and column names. We also create metadata about file processing, such as when files were processed, their sizes, where they were saved and what data files were combined to produce the resulting file.

There are several ways to store metadata. Some file formats such as NASA-AMES used by ACTRIS, or hierarchical data format (HDF) files store it in the data file itself. For us this would result in duplication of the metadata, and it is not well suited for file processing metadata. We store the metadata as database tables and link to files, as necessary. It is possible to export metadata from the databases to file formats when required by workflows."

**Line 180: Figure description has to include all abbreviations used in the figure**

We added expanded the abbreviations (https, S3, SSH, SCP, SQL, InfluxQL) in the caption and explained the terms.

**Conclusions, line 382:**

**This makes it useful for measurements that are not controlled by centralized solutions**

**Add: or it can be used as a backup of the data owner in parallel with data transfer to centralized networks**

We added the suggestion.

Conclusions, line 385.

Your current text:

We show in two case studies how continuous visualizations of the data and metadata, such as instrument diagnostics and datafile availability, can help quickly spot interesting phenomena and abnormal situations in the measurements.

Suggestion:

We demonstrated with two case studies how continuous visualizations of data and metadata, such as device diagnostics and data file availability, can help to quickly identify interesting phenomena and abnormal situations in measurements

We implemented the suggestion, with a small change: "help quickly" instead of "help to quickly".